# The Effects of Pen Size and Design, Bedding, Rooting Material and Ambient Factors on Pen and Pig Cleanliness and Air Quality in Fattening Pig Houses

**DOI:** 10.3390/ani12121580

**Published:** 2022-06-19

**Authors:** Marko Ocepek, Inger Lise Andersen

**Affiliations:** Department of Animal and Aquacultural Sciences, Faculty of Biosciences, Norwegian University of Life Sciences, 1432 Aas, Norway; inger-lise.andersen@nmbu.no

**Keywords:** pig and pen cleanliness, ammonia concentration, pen design, rooting material, temperature, air velocity

## Abstract

**Simple Summary:**

Inappropriate eliminating on a resting area has a negative effect on the environment, the cleanliness of pigs and pens, and can impair farm productivity. There are several factors that can affect pig eliminative behaviour. The primary aim of this survey was to investigate which factors related to the physical environment have the strongest effects on pig and pen cleanliness and ammonia concentration. Secondly, we wanted to assess the relationship between ambient temperature, air velocity and cleanliness of the pen and the pigs and ammonia concentration. Data were collected from 87 pig farms (*n* = 5769 pigs; *n* = 643 pens). The larger the eliminative area and resting area, the cleaner were the pigs. The eliminating area could have open partitions and be placed in the back of the pen. The resting area with a large amount of litter on the floor and use of straw as rooting material increased the cleanliness of this area. A more optimal pen design, such as that described in the present study, can reduce the workload for the farmers (cleaning), improve air quality, and lower the environmental footprint.

**Abstract:**

There are several environmental and ambient factors that can affect pig eliminative behaviour. The aim of this survey was to investigate factors related to the physical and ambient environment that have the strongest effects on pig and pen cleanliness and ammonia concentration. Data were collected from 87 pig farms and analysed using mixed (ammonia concentration) or generalized linear (pen and pig cleanliness) model in SAS. The pen was cleaner when pen partitions were open compared to closed (*p* = 0.010) and increased with increasing amount of litter (*p* = 0.002), using straw (*p* = 0.002) as rooting material. Pig cleanliness was higher when pen partitions in the eliminative area were open compared to closed (*p* = 0.007) and increased with increasing space per pig in the resting area (*p* < 0.001), with decreasing temperature (*p* < 0.001), and lowering of air velocity (*p* = 0.003). Other factors that increased cleanliness was using straw as rooting material (*p* = 0.028) and increasing amount of litter in the resting area (*p* = 0.002). Ammonia concentration was reduced with increasing floor space in the eliminative area (*p* < 0.001) and increasing amount of litter (*p* = 0.006). Our results pinpoint physical and ambient conditions affecting pen and pig cleanliness and air quality.

## 1. Introduction

Understanding pig eliminative (urination, defecation) behavior is important for welfare and environmentally friendly pig production. Pigs are by nature clean animals, and they prefer to differentiate between designated areas for resting, feeding and eliminating [1,2]. In man-made environments, space is limited, and pens are often not designed to meet behavioural needs. Under such condition, pigs begin to eliminate in the resting area and this can have a negative effect on the cleanliness of pens and pigs [3,4], air quality [5,6], human and pig health [7]. Finally, this can impair farm productivity.

According to the Norwegian animal welfare regulations, pigs should have access to a comfortable, dry, clean, and draft-free lying area (solid floor with litter or deep bedding) [8]. The pen should consist of a separate lying and an eliminating (slatted floor) area. It is stated that the lying area has to be covered with “sufficient amount” of litter and “large enough for all of the pigs to rest simultaneously”. The other part of the pen can be used for fouling. Pigs should have access to rooting material the whole time. Material such as straw, hay, sawdust, peat, wood shavings or a mixture of these can be used. At the end of the fattening period (110 kg) it is stated that each pig should have access to at least 0.8 m^2^ area in total. Finally, ammonia concentration should not exceed 20 ppm, but is it recommended to be less than 10 ppm. In pig lying area air velocity is preferred to be less than 0.2 m/s, within the comfort thermal zone [8].

Pen size and design are important factors in pig production. Appropriate eliminative behaviour reduces over time due to increased density [9]. Hillmann et al. [10] reported that with reducing space allowance, pigs were lying even more in the area for elimination. Other pen design factors such as pen partitions and position of eliminative area can ameliorate this negative trend. Pigs prefer to eliminate in corners or near pen partitions (marking), especially in open (slatted) ones (increased air velocity), but this increases the risk of attacks from neighboring pigs [11]. Placement of proper partitions, while understanding pig needs, could improve the use and functionality of designated areas (eliminating and resting). There are some contradictory results about the position of the eliminative area [12,13], and thus encouraging use of the elimination area could depend on location in the pen (placed in the front, in the back end, or on both sides). A combination of factors such as pen size (slatted floor and solid floor per pig) and design (pen partition, position of eliminative area) could have an impact on elimination behaviour in pigs, especially while relating them to other factors.

It is crucial to ensure comfort on the resting area [14,15]. Provision of bedding material (litter or deep bedding) could improve comfort, but with increased eliminative behavior in this area, this could lead to contamination of the litter or bedding material, poor pig and pen hygiene. How much litter is “sufficient” on the resting area is not well documented. Temple et al. [16] visited 91 farms and reported that deep bedding systems had poorer hygiene than conventional pens with solid floor. Pigs should also have access to rooting material the whole time. Rooting material is important for welfare [15,17,18,19,20,21], but also raises economical (cost of rooting materials [22]) and practical issues (methane production [15,23]). While straw is commonly used material for rooting, there is some indication that peat, maize silage, branches, sawdust may not be inferior to straw. Pedersen et al. [18] observed that fattening pigs preferred peat and branches over straw, while Ocepek et al. [15] reported that maize silage can replace straw. Studnitz et al. [21] concluded that peat, sawdust, silage, mushroom compost, sand, wood shavings, branches and beets all ranked above straw. However, in many circumstances, straw is more easily available for farmers. In addition to type of material or combination of materials, the frequency is also important. Apart from Olsson et al. [24], that observed effects of different rooting material on lowering ammonia emission in outdoor organic pig production, reports on the relative impact of different materials on eliminative behaviour in conventional indoor rearing are lacking. How the amount of bedding, different rooting materials, or frequency of provision can affect pigs’ eliminating behaviour requires further investigation.

Climatic conditions may also influence eliminating behavior to a great extent. At higher ambient temperatures (≥20 °C), fattening pigs search for a cooler resting area. Under such condition, pigs start to prefer slatted over the solid floor for resting as over the slatted floor temperature lowered by 4 °C [13]. At the same time, pigs can reduce level of elimination on eliminative area, for not having enough space (other pigs lying) or intentionally eliminate in the resting area to wallow in own excretion [13] in order to reduce floor temperature and cool body surface [25,26]. Air velocity presented is mainly related to thermo-neutral conditions and can be increased with increasing temperature. As both ambient factors have huge impact on eliminative behavior, they need to be analyzed systematically together with the other factors mentioned above.

The primary aim of this survey was to investigate which factors related to the physical environment have the strongest effects on pig and pen cleanliness and ammonia concentration. In particular, we wanted to focus on effects of pen design (pen partitions, space per pig on solid and slatted floor), rooting material (type and provision frequency) and amount of litter in the resting area. Secondly, we wanted to assess the relationship between ambient temperature, air velocity over resting area and cleanliness of pigs and pens as well as ammonia concentration.

## 2. Materials and Methods

### 2.1. Farm Selection and Study Design

Eighty-seven fattening pig farms were visited, meeting the following criteria: (1) had most common fattening pig breed in Norway (offspring of TN70 crossbreed sows from Norsvin Landrace and Topigs Norsvin Z-line (Yorkshire) and inseminated with Norsvin Duroc boar semen); (2) distributed between all four Norwegian regions (East, *n* = 27; West, *n* = 20; and Middle, *n* = 20; North, *n* = 20); and (3) fattening pig producers owning Nortura SA (*n* = 16,300). Pig farms were visited by trained Nortura regional advisors, or the researcher involved within one year (January–December 2020). Registrations were conducted once between 10:00 and 12:00 h during the last three week before slaughtering, when pigs in the pen were at highest density. Eight pens per farm were randomly selected, with at least one pen in between not used. If the farm had fewer than eight pens, all of the pens were included in the study. All pigs in the pen were scored, but if the number of the pigs exceeded 15, we randomly marked and scored 10 pigs in the pen. We organized three observation training and calibration trials (testing protocols in two farms per trial) for Nortura advisors to ascertain that interpretation of the protocol was the same for all advisors.

### 2.2. On Farm Registration

A scoring system for pen and pig cleanliness was developed in a pilot study on 10 commercial farms.

*Pen cleanliness.* The presence of manure (urine and/or feces) on the solid floor (with litter) was visually assessed by standing in front of the pen using a scale from 1 to 3. 1 (dirty): more than 40% of the floor covered with manure); 2 (moderate dirty): between 10 ≥ 40% of the floor covered with manure; 3 (clean): less than 10% of the floor soiled.

*Pig cleanliness.* The assessment was performed while being inside the pen, when pigs were in standing position. The presence of manure on pig body (urine and/or feces) was visually assessed on the whole-body surface using a scale from 1 to 3. 1 (dirty): more than 40% of the body is surface is soiled); 2 (moderate dirty): between 10 ≤ 40% of the body surface is soiled; 3 (clean): less than 10% of the body surface is soiled.

*Pen size.* The inside length and width of the solid floor (resting area) was measured as well as the inside length and width of the slatted floor (eliminating area).

*Pen design*. Pen partitions (open (slatted), partly open (lower part of the partition is closed and upper part is slatted, that lying pig could not be disturbed, but while standing pig could have contact with neighbour pigs), or closed (completely solid partition)) at both solid and slatted floor, and the location of the slatted area (in front (toward the corridor), back (toward the wall)) or on both sides was registered. Even though there was variability within the pen size per pig and design (*n* = 70 types), the most traditional shape and size is shown in Figure 1.

*Amount of the litter on the solid floor.* Amount of litter on the solid floor was assessed using 1 to 5 scale (Table 1). 1: Litter not used; 2: Small amount-little litter over the entire solid floor area, with visible larger parts of the floor through the litter; 3: Moderate amount-litter distributed over the entire solid floor, with visible smaller parts of the floor through the litter; 4: Large amount-litter distributed over the entire solid floor, with no visible floor through the litter; 5: Deep bedding-deep bedding used as litter, provided continuously).

*Rooting material.* Discussed with pig producers. We noted rooting material type and provision frequency of it during the current batch. The pigs were provided different rooting material on farms, and as more than one material was provided to 86% of the pens, each type of rooting material was categorized into two classes (not provided = class 1; provided = class 2). Rooting material was removed at first pen cleaning. Provision frequency of rooting material varied from weekly, daily, twice a daily, to more than twice a daily across farms.

*Temperature, air velocity and ammonia concentration measurements.* Temperature, air velocity (VELOCICALC, temperature and air velocity meter, model 9515, VelociCalc Air Velocity Meter 9515 | TSI) and ammonia concentration (GfG Micro IV, GfG Instruments, https://www.gasdetectorshop.com/Micro-IV-GfG-Single-Gas-Monitor-1418-p/gfgmicroiv.htm, accessed on 1 January 2020) were measured 20–30 cm above the solid floor area (pig height in the resting area) in three pens across the room (first, middle and in the last pen).

### 2.3. Statistical Analyses

Descriptive statistics were presented with means and SE. Statistical analyses were conducted using SAS 9.4 statistical software (SAS Institute Inc., Cary, NC, USA). The effects of pen design (pen partitions (open, partly open and closed), space per pig on the solid (m^2^) and on the slatted floor (m^2^), location of the slatted floor (in the front, back, and both sides) on pen cleanliness was analysed using the GLIMMIX procedure with multinomial distribution. Regarding pig cleanliness, a GLIMMIX procedure with binomial response distribution was used, and ammonia concentration was analysed using a MIXED model (Proc Mixed, due to normally distributed residuals). Amount of litter in the resting area (little, moderate, large, deep bedding), rooting material type (chopped or long straw, silage, hey, newspaper, wood shavings; each one as separate class variable; not provided = class 1; provided = class 2) and rooting material provision frequency (weekly, daily, twice daily, more than twice daily), were included in the model as fixed effects (class variables). Ambient temperature (°C) and air velocity (m/s) over resting area were continues variables in the model. Farm ID was specified as a random effect, as there were repeated measures per farm. Pairwise means comparisons were based on differences in least squares means with Tukey adjustment for multiple comparisons.

## 3. Results

### 3.1. Descriptive Data

The data collected from 87 farms contained information on 5769 individual pigs from 634 pens.

*P**en and pig cleanliness.* Pen and pig cleanliness was scored (Table 2). Pig cleanliness that refers only to score 3 (less than 10% of the body surface soiled) was used. Presented as the percentage of the pigs with less than 10% of the body surface soiled in the pen.

*Pen size per pig.* Pen slatted floor and solid floor per pig was calculated (Table 2).

*Pen design.* Pen design such as pen partition by the solid and the slatted floor in the pens as well as location of the slatted area in the pens was registered (Table 3). There were no collected data on open pen partition by the solid floor.

*Amount of the litter on the solid floor*. Amount of litter on the solid floor was noted (Table 4). There was no pen without the litter on the solid floor (class 1).

*Rooting material.* Type of rooting material and provision frequency was collected (Table 4).

*Temperature, air velocity and ammonia concentration measurements.* Data on temperature, air velocity and ammonia concentration were presented as mean value per room (Table 5).

### 3.2. Effect of Pen Size Per Pig and Design

Cleanliness of the pen and the pig as well as ammonia concentration was significantly affected by pen design. Pen cleanliness increased with increasing slatted floor per pig (F_1472_ = 4.9, *p* = 0.027; Mean ± SE: dirty = 0.35 ± 0.01 m^2^/pig; moderate dirty = 0.38 ± 0.03 m^2^/pig clean = 0.41 ± 0.02 m^2^/pig). Pen with open partitions on the slatted floor were significantly cleaner than pens with closed pen partitions, with partly open being intermediate (F_2472_ = 3.8, *p* = 0.010, Figure 2A). Pen cleanliness was highest in the pens with slatted floor in the back, followed by with slatted floor on both sides and lowest in the pens with slatted floor in the front (F_2472_ = 5.3, *p* < 0.001, Figure 2B). Pig cleanliness increased with increasing solid floor per pig (F_1472_ = 19.7, *p* < 0.001, Figure 3A), and was greatest in the pens with open pen partitions and lowest in the pens with closed pen partitions of the slatted floor (F_2472_ = 4.1, *p* = 0.007, Figure 2C). Ammonia concentration was greatest in the pens with slatted floor in the back and lowest in the pens with slatted floor in the front, with pens on both sides being intermediate (F_2472_ = 4.3, *p* = 0.008; Figure 2D). The ammonia concentration increased with increasing solid floor per pig (F_1472_ = 69.3, *p* < 0.001; Figure 3B) and declined with increasing slatted floor per pig (F_1472_ = 100.3, *p* < 0.001; Figure 3C).

### 3.3. Effect of Litter Amount on the Solid Floor

Litter amount on the solid floor affected the pen and the pig cleanliness as well as ammonia concentration. The pen (F_3472_ = 5.1, *p* = 0.002, Table 6) and the pig (F_3472_ = 5.0, *p* = 0.002, Table 6) cleanliness was higher in the pens with large amount of litter, followed by the moderate, and lowest with little and deep bedding, respectively. Ammonia concentration was highest in the deep bedding pens and lowest in the pens with large amount of litter (F_3472_ = 25.0, *p* < 0.001, Table 6).

### 3.4. Effect of Different Rooting Materials on the Solid Floor

Rooting material on the solid floor affected pen and pig cleanliness. Chopped (F_1472_ = 5.8, *p* = 0.017, Figure 4A), long straw (F_1472_ = 9.6, *p* = 0.002, Figure 4B) and hay (F_1472_ = 4.5, *p* = 0.035, Figure 4C) improved pen cleanliness. Similarly, pig cleanliness was higher in the pens with chopped (F_1472_ = 4.9, *p* = 0.028, Figure 4D) or long straw (F_1472_ = 4.3, *p* = 0.038, Figure 4E) on the solid floor.

### 3.5. Effect of Rooting Material Provision Frequency

Pen cleanliness was highest with provision of rooting material once (mean ± SE, 2.4 ± 0.0), twice (mean ± SE, 2.5 ± 0.0) and more than twice (mean ± SE, 2.6 ± 0.1) daily, and lowest in weekly provision of rooting material (mean ± SE, 1.8 ± 0.2; F_3472_ = 3.3; *p* = 0.022). There was no significant effect of rooting material provision frequency on pig cleanliness (F_3472_ = 1.0; *p* = 0.386) or ammonia concentration (F_3472_ = 1.1; *p* = 0.376).

### 3.6. Effect of Temperature and Air Velocity

Pig cleanliness decreased with increasing temperature (F_1472_ = 28.9, *p* < 0.001, Figure 5A) and air velocity (F_1472_ = 8.8, *p* = 0.003, Figure 5B). Ammonia concentration increased with increasing air velocity (F_1472_ = 115.2, *p* < 0.001, Figure 5C).

## 4. Discussion

Our study demonstrates that pen and pig cleanliness as well as ammonia concentration were affected by multiple on-farm factors, such as pen design, bedding material, rooting material, and ambient climate in a systematic way. Pen cleanliness was influenced by space per pig in the eliminative area, pen partitions of the resting area, the slatted area location, amount of the bedding material, type of rooting material, rooting material provision frequency. Pig cleanliness was affected by solid floor area per pig and not by the size of the slatted floor. Other factors influencing pig cleanliness were presence and design of partitions, amount of bedding material, type of rooting material, air temperature, and air velocity. Solid and slatted floor area per pig, slatted floor location, amount of bedding material and air velocity all affected ammonia concentration.

### 4.1. Pen Size Per Pig

In accordance with Aarnink et al. [12], we found that an increased eliminative area resulted in pens (resting area) becoming cleaner and a reduction in ammonia concentration. Even more, our results showed that cleanliness of the resting area could be achieved by increasing the eliminative area by 17%, and that the resting area was cleanest when eliminative area was at least 0.41 m^2^/pig. The data indicates that eliminative area needs to be large enough for several pigs to eliminate simultaneously in this designated area. Eliminating in this area reduces ammonia concentration and emissions as on the slatted floor, urine drains into the pit. [12]. If there is not enough space in eliminating area, pigs would likely to choose resting area and this would lead to increased ammonia concentration and emission [12]. Our data show that pig cleanliness and ammonia concentration increases with increasing solid floor area. The likelihood of pigs eliminate in resting areas increases with more space provided. It is also crucial that the resting area is large enough. How much of resting area is required per pig and whether this is proportional to the amount of solid/slatted floor per pig requires further investigation. Ocepek and Škorjanc [27] reported that if pigs elimination on the solid floor increased by 1%, this resulted in an ammonia volatilization escalation by 0.3 g/pig. As ammonia concentration and emission is produced after elimination on the solid floor, this cannot be sustained or prevented any longer [28]. Larger resting area gives pigs opportunity to choose place they would lie on, stay cleaner and thus, pigs may benefit from having multiple lying areas in different locations. Farmers in current study provided 60% more space per pig than described in legislation (1.3 m^2^ vs. 0.8 m^2^). However, we also found that a larger resting area led to higher ammonia concentration. Again, this underpins the importance that the design and location of different functions of the pen needs to meet behavioral needs of the pigs, and that existing pens are not designed in an optimal way. Resting area need to be large enough for pigs to move around and to eliminate on designated areas, while other pigs are resting. Free “walking paths (without other pigs being in the way)” between functional areas could be taken under consideration. Nonetheless, our results documented that the amount of slatted and solid floor per pig in the current study was too low and could be improved.

### 4.2. Pen Design

Another possibility to improve pen and pig cleanliness and reduce ammonia concentration would be by improving eliminative area with focus on location and pen partitions around it. We found that the best position of the eliminating area was in the back of the pen since such pens were cleanest and had lowest ammonia concentration. Previous studies found that placing the eliminative areas in both sides compared to only in the back, would also reduce ammonia concentration and emission [12,13], but these studies did not test the location of the eliminative area in a systematic way. To our knowledge, our study is the first to document that placing eliminative area in the back of the pen, where most of the fouling occurs, could improve pen cleanliness and lower ammonia concentration compared to other locations. This part of the pen is exposed to direct outdoor climatic factors, such as warmth (sun), cold on wintertime, windows, etc. Placing eliminative area only in one place would reduce housing costs for the farmers (only one slurry pit needed; reduced need for heating). Open pen partitions on the slatted floor rather than partly or fully closed, increased pen and pig cleanliness. The eliminating area should be less attractive, only to perform elimination. Open pen partitions provide additional airflow from the sides, making this area less attractive for other activities such as resting. Furthermore, open partitions may stimulate pigs to mark them (increased eliminating), thus making territorial borders with neighbouring pigs [29]. In contrary, pen partitions of the resting area should be closed to protect pigs from uncontrolled air flow and from neighbouring pigs [11]. Our data also showed that there was no difference between fully closed or partly open (upper part) pen partitions in the resting area, meaning that both types could be used.

### 4.3. Effect of Litter Amount on the Solid Floor and Rooting Material

Interestingly, we found that with an increasing amount litter on the resting area from little to large amount, pen and pig cleanliness increased and ammonia concentration declined. This suggests that a large amount of litter in the resting area make the pigs preserve this area and eliminate less. Indeed, having a comfortable resting area is of great importance for pigs, especially if they spend 80% of their life resting [15]. An increased resting area in combination with a large amount of litter on the floor, would most likely result in a cleaner resting area overall.

Going from a larger amount of litter on the resting area to the systems of deep bedding, one might predict that this would improve pen and pig cleanliness and ammonia concentration. However, this was not the case. Farmers have more problems managing deep bedding systems then traditional pens. In deep bedding, provision of additional straw should be carried out weekly, and better ventilation systems are required, especially during warm summer days. In previous research, deep bedding systems has resulted in higher welfare status in terms of less tail and ear biting and body lesions [19]. However, our results showed the pigs were often dirtier and ammonia concentration was higher in some farms with straw bedding than in traditional pens. In our view, an improvement could be achieved by having straw bedding resting area, combine with additional access to eliminative area.

In addition to provision of litter in the resting area, provision (at least once daily) of rooting material also had a positive impact on pen and pig cleanliness. This effect was greatest with straw, regardless of being chopped or long. As rooting material is usually provided on the solid resting area, this could potentially increase the value of this area for the pigs and thus reduce the risk for eliminating in this area.

### 4.4. Temperature and Air Velocity

As expected, we documented that with increasing temperature, pig cleanliness decreased. At higher temperatures (≥20 °C), pigs need areas for cooling. Under such condition, pigs would prefer to lie and expose most of the body to cooling area. They would avoid close contact to pen mates and most likely try to wallow in excretion in eliminating and/or resting areas, which leads to lowered body cleanliness, as we documented. They perform this type of behavior to be able to cool down by being wet. Although, air velocity should not be higher than 0.2 m/s, but at higher temperature, increased air flow over resting area could help pigs overcome such heat stress. In addition, sprinkle system could help pigs to cope with high temperature during hot summer days. Sprinkle system can be installed over the eliminative area, making this area even wetter.

We have tested several physical and ambient on-farm factors affecting pen and pig cleanliness and ammonia concentration. While we showed that slatted floor should increase, increased resting area have both positive and negative effects (improving pig cleanliness, as well as increasing ammonia concentration). This can be even more problematic during warm summer days, with sub-optimal air velocity over resting area, especially in deep bedding systems. More systematic studies including ambient factors and its effect on pig behavior, use of pens is still needed to find compromise between cleanliness and satisfying pig needs. By use of new digital tool and algorithms, we could detect pen fouling in real time, obtaining more knowledge about when and why this happens [30].

## 5. Conclusions

The larger the eliminative area and resting area, the cleaner the pigs and pens. However, larger areas resulted in higher ammonia concentration which could be mitigated by open or partly-open partitions in the eliminative area at the back of the pen. In addition, provision of large amounts of litter on the solid floor and using straw as rooting material increased pig and pen cleanliness whereas inappropriate use of the pen areas for elimination decreased cleanliness. In this study, we have identified links between housing factors that affect pig and pen cleanliness. Future research will investigate direct links and improvements to pig housing that minimize risks that negatively impact the health and overall welfare of commercially farmed pigs.

## Figures and Tables

**Figure 1 animals-12-01580-f001:**
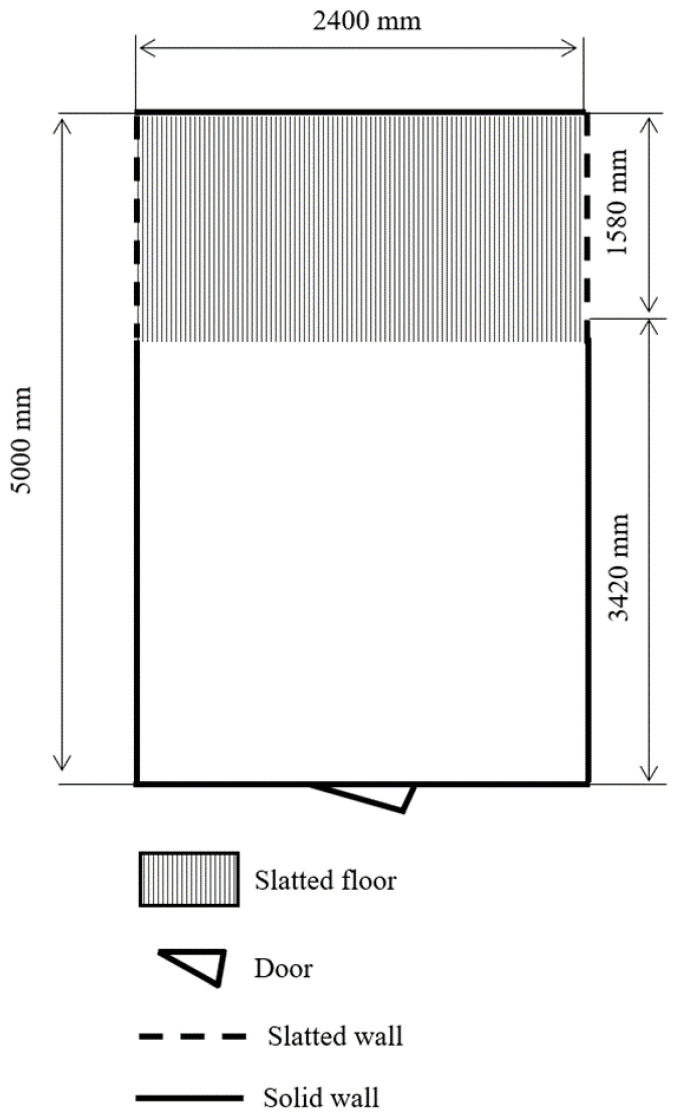
Pen layout.

**Figure 2 animals-12-01580-f002:**
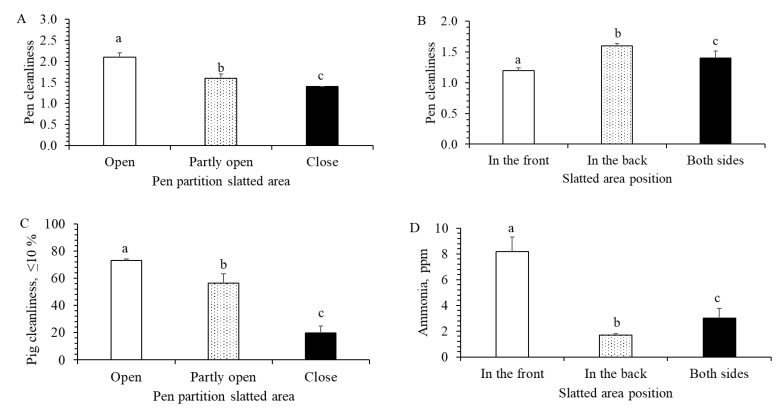
Effects of pen design on pen and pig cleanliness and ammonia concentration. (**A**) Pen cleanliness and pen partition of slatted floor (F_3472_ = 3.8; *p* = 0.010). (**B**) Pen cleanliness and slatted area position (F_3472_ = 5.3; *p* < 0.010). (**C**) Pig cleanliness and pen partition slatted area (F_3472_ = 4.1; *p* = 0.007). (**D**) Ammonia concentration and slatted area position (F_3472_ = 4.3; *p* = 0.008). a, b, c = means with different superscripts are significantly different (*p* < 0.05).

**Figure 3 animals-12-01580-f003:**
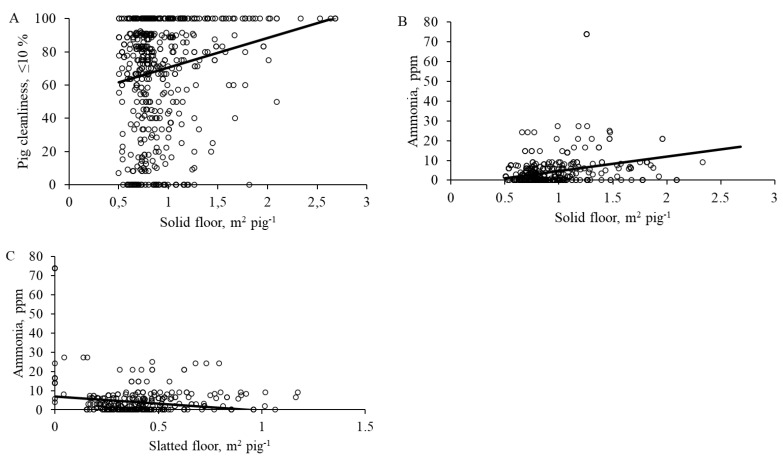
Effect of pen size on pig cleanliness and ammonia concentration. (**A**) Pig cleanliness and solid floor area per pig (F_1472_ = 19.7; *p* < 0.001). (**B**) Ammonia concentration and solid floor area (F_1472_ = 69.3; *p* < 0.001). (**C**) Ammonia concentration and slatted floor area (F_1472_ = 100.3; *p* < 0.001).

**Figure 4 animals-12-01580-f004:**
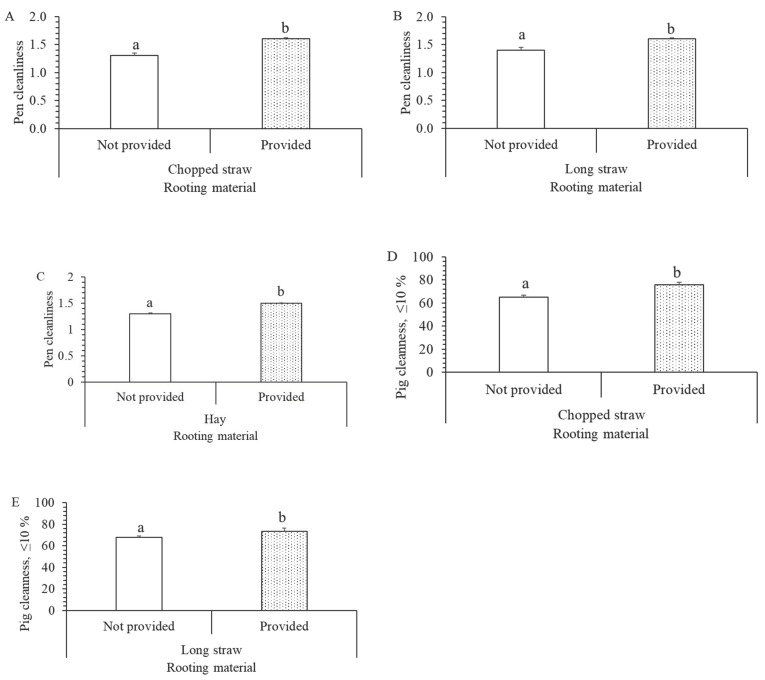
Effect of different rooting materials on pen, pig cleanliness and ammonia concentration. (**A**) Pen cleanliness and chopped straw (F_1472_ = 5.8; *p* = 0.017). (**B**) Pen cleanliness and long straw (F_1472_ = 9.6; *p* = 0.002). (**C**) Pen cleanliness and hay (F_1472_ = 4.5; *p* = 0.035). (**D**) Pig cleanliness and chopped straw (F_1472_ = 4.9; *p* = 0.028). (**E**) Pig cleanliness and long straw (F_1472_ = 4.3; *p* = 0.038). a, b = means with different superscripts are significantly different (*p* < 0.05).

**Figure 5 animals-12-01580-f005:**
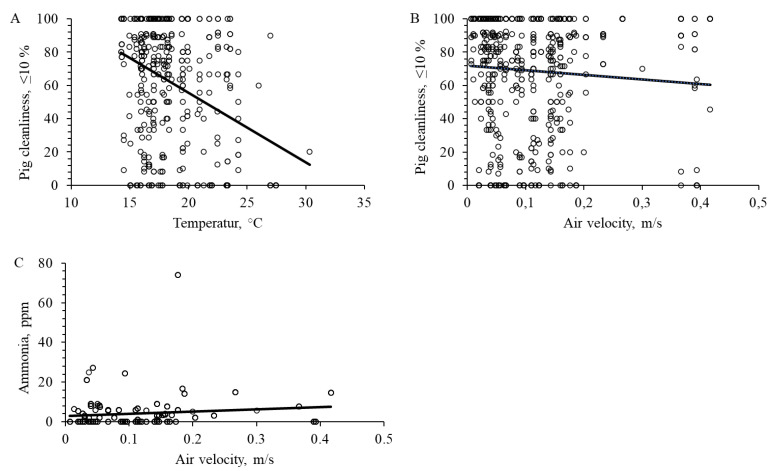
Effect of ambient temperature and air velocity on pen and pig cleanliness, and ammonia concentration. (**A**) Pig cleanliness and temperature (F_1472_ = 28.9; *p* < 0.001). (**B**) Pig cleanliness and air velocity (F_1472_ = 8.8; *p* = 0.003). (**C**) Ammonia concentration and air velocity (F_1472_ = 115.2; *p* < 0.001).

**Table 1 animals-12-01580-t001:** Amount of the litter on the solid floor.

Litter Not Used	Small Amount	Moderate Amount	Large Amount	Deep Bedding
	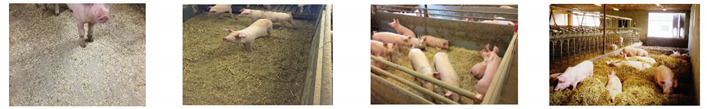

**Table 2 animals-12-01580-t002:** Descriptive data for pen and pig cleanliness and pen size.

	Mean ± SE	Range
Pen cleanliness, 1–3	1.5 ± 0.0	1–3
Pig cleanliness, %	69.4 ± 1.4%	0–100
Pen size:		
Slatted floor per pig, m^2^	0.4 ± 0.0	0–1.8
Solid floor per pig, m^2^	0.9 ±0.0	0.5–2.7

**Table 3 animals-12-01580-t003:** Descriptive data for pen design.

	Pens, %
Pen partition by the solid floor:	
Open	0
Partly open	31
Closed	69
Pen partition by the slatted floor:	
Open	88
Partly open	7
Closed	5
Location of the slatted area:	
In the front	55
Toward the wall	40
Both sides	5

**Table 4 animals-12-01580-t004:** Descriptive data for litter amount and rooting material.

	Pens, %
Litter amount:	
Litter not used	0
Small amount	37
Moderate amount	43
Large amount	17
Deep bedding	3
Type of rooting material:	
Chopped straw	28
Long straw	22
Silage	41
Hay	39
Newspaper	41
Wood shaving	21
Provision frequency of rooting material:	
Weekly	2
Daily	49
Twice a daily	46
More than twice a daily	3

**Table 5 animals-12-01580-t005:** Descriptive data for temperature, air velocity, and ammonia concentration.

	Mean ± SE	Range
Temperature, °C	17.9 ± 0.1	11–30
Air velocity, m/s	0.1 ± 0.0	0.0–0.42
Ammonia concentration, ppm	4.1 ± 0.4	0–74

**Table 6 animals-12-01580-t006:** Effect of litter amount on pen, pig cleanliness and ammonia concentration.

	Litter Amount (Mean ± SE)			
	Little	Moderate	Large	Deep Bedding
Pen cleanliness	2.3 ± 0.0	2.5 ± 0.0	2.8 ± 0.0	2.2 ± 0.1
Pig cleanliness, ≤10%	64.1 ± 2.4	70.5 ± 2.1	80.8 ± 2.5	49.0 ± 9.1
Ammonia concentration, ppm	3.2 ± 0.3	3.4 ± 0.4	1.9 ± 0.7	24.3 ± 5.4

## Data Availability

Not applicable.

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
