# Peer review of "The Effects of Pen Size and Design, Bedding, Rooting Material and Ambient Factors on Pen and Pig Cleanliness and Air Quality in Fattening Pig Houses"

_animals, 2022, doi:10.3390/ani12121580_

Round 1
Reviewer 1 Report
L8: on 'a' resting area
L11: remove 'that'
L13: commas around as well as
Line 14: syntax and tense - was, not is
L15: The eliminating area.
L16: temper - quality of the resting area had an effect on usage etc. How did you measure value? Just temper this sentence a bit and return to the independent variables - you can say this simply without overstating (or consider deleting).
L26: The pen
L34: I find 'great importance' difficult. Possibly temper this bit.
L41: I don't understand this sentence. 'but also to? cleaning the pens'. Start new sentence after production. Ventilation sentence is out of place here.
L52: optimal? consistent - if you say optimal I want to know what that is
L52 The pen
L57: expand on 'fattening period' for those of us who aren't in the know - 0.8m2 is not a lot of space at all (guinea pigs have more!) expand (should have access to x space for lying? or do you mean that 0.8m2 per pig designates the overall pen size? Please clarify
L59: dunging, fouling, eliminating - stay consistent
L60: put in previous paragraph as you are talking about dunging now? I'm getting lost.
L63: change proper to appropriate?
L67: open ones ? clarify parentheses too
from L69: restructure and eliminate the question - saying the question remains I find annoying - restructure this "Encouraging usage of the elimination are could depend on location in the pen...." rather than present the info as a question answer, just state the issue.
L73 would make a good intro to L65 onwards (except for the great important bit). I need more structure here, space, partitioning, location, air quality - present logically and concisely without questions.
L77: the comfort + ref
L81: citation style?
82: expunge 'great importance' :)
L86: conventional pens with a concrete floor?
L88: wording: link between provision of bedding and rooting material and elimination behaviour requires further investigation?
L93: finish the thought, thus the elimination area moves? or does elimination become ubiquitous? (I wonder if there is stimulus control at work here - the type of floor signals the usage and when the temperature rises, then stimulus controls decreases). If this is research, maybe state that as it reads as if slatted floor decreases temp by 4 degrees everywhere.
L96, remove that
I have reached the end of the introduction. I have learned a bit about what you are going to talk about but you haven't mentioned amount of litter in in the intro text or research that has compared straw or other rooting material (I know Pedersen et al., 2005, and Holm et al., 2008) have done some work on preferences for rooting material. There should also be expectations or hypotheses for your research based upon the literature you cite in the intro. I think you need to improve the structure a bit and provide a paragraph (ish) for each of your test variables with what research has been done for that variable. As the reader, I need to have background info.
Method:
L11 - n=16.3?
L112: once
L113 when pig pens were at the highest density
L115 fewer, not less
L116 - how did you choose which pigs to score?
L123 - is this with litter? or without?
L124 The assessment, 'when pigs were standing'.
How did you come up with the scale? was it based on published research?
Data collection: ? but it's results. I would put the following section as a table in the results and have one paragraph with the main trends.
Pen design section: simplify, and watch grammar. Could say that pen size and design was noted. I find this hard to digest - maybe refer to a table in the results for design types?
Amount of litter - not very specific - did you do interobserver reliability? Who did the measuring? How did you remain consistent? Did you take photos and assess? what is deep bedding - again, it isn't specified enough in the intro for to me to understand what this might have looked like.
Rooting - mention assessment, not results.
L147 - how? this is the method - you need to specify how you measured this variable so it is replicable.
L154 - what do you mean 'on the pen level'? 2.3 - these are results? and it's in the method? move 2.3 to results.
L156 - mean doesn't really mean anything if you've provided percentages and there are only three options.
Table 1 - delete - graph with your statistics presented in the text is adequate and much clearer.
L161 - wording. You don't need the range if you have the standard error? or SD? say which.
L161 - I'm not sure how you got to this calculation. 57% of pens were clean, 68% of pigs were clean - explain how you got to 69% please. Make sure the stats you provide are meaningful - I think just percentages is useful to me.
L163 across farms? so number of pigs divided by slatted floor area? I think a table would help here. Put a column of design options, then the percentage of each. Relate to pen design section of method.
Line 172 - bedding or litter? be consistent. what is class 1?
174 hay
L176 - this explanation of class needs to go earlier.
L177 - mention in method that you surveyed this info.
Ammonia? Air concentration?
L194 - farm? What type of measure is this? categories? which?
L182 - concisely put - good!
L201 on - 2dp for SE
L203 - intermediate? What does this mean?
For this paragraph, put the statistics in for each comparison. This is meaningful. Figure 1 - great! Figure 2 - look at spelling. 2A is showing the pigs that scored a 3 for being clean - correct? But then how is the y axis up to 100? explain.
L232 - just report trends - L235 is superfluous. Put stats in the text where relevant - not in the table.
L245 hay
Table 3 - make it like table 2 and put stats in the text
figure 3 - combine ABC, and DE instead of having 5 graphs with a not provided bar.
Good - I'm not sure if the journal requires it but you may need to include effect sizes - these can allude to the most meaningful of your results.
Table 4 put stats in the text and have averages etc in the table.
Figure 4 - watch spelling. I like these .
Thoughts: you mention measuring the size of the pens but you did not analyse this data?
L284 per
L286 strongest affected by
L286, presence and design of partitions
L295 delete gradually (you didn't measure time)
Discussion - first paragraph. Good compare contrast. overall more space (I think every captive animal would agree with you). You mention a larger resting area with enough space for all to rest is needed, but this increases ammonia concentration - so is it the size of the elimination area? Line 300 what data specifically? move this sentence. Therefore, it seems to me that although a larger rest area would appear beneficial for what you have said (behavioural needs) it actually results in a problem - so how might you go about managing the size/or design of the pen and in particular the eliminative area - you talk about this in the next paragraph but you need to set it up with why size may not be the resolution.
L318 - does size of the eliminative area/slatted area matter? either put something about this before now as you've already set up size to be a factor. Then talk about other factors.
L321 - always? not mentioned this before - do you mean air velocity? Include something about farm design in the method to give scope for this sentence.
L325 - tell me more (in the intro). I've alluded to this already - but the slatted floor should be signal enough but why are pigs not using it? This would be a paragraph in the intro. Such as more about L326 - this is interesting but I needed to know earlier
L339 - good set up
Line 340 - new paragraph
L342, in previous research, provision of straw has resulted in
L344 - your study? make this clear (Table 1 with all variables and percentages would give the reader this info)
L345 - but you just said straw resulted in higher ammonia levels? What about the frequency of replacing straw - does this have an effect?
L350 remove emotion 'rather surprising'
L350 it seems that straw has multiple functions but also requires a lot more effort to manage. It might increase the 'value' (although I'm convinced on this word use as you didn't measure this) but it also increases ammonia.
L355 - don't assume :) expect?
L360 - I think you could go further here (you've mentioned it in the intro but maybe make more of a point of it). What about providing more air velocity during that time in the resting area - expand on your data and what you have learned. Watch grammar here. A sprinkler system etc - but wouldn't this make one huge mess??
L364 - this is not what you have said earlier - larger area = higher ammonia. Watch grammar.
I'm all for providing a recommendation, but I think you should temper this and say that with advantages in some areas (e.g., semi-open partitions increasing air flow, there are disadvantages which affect usage of pen areas inappropriately causing issues of cleanliness and hygiene, and thus can negatively impact welfare, particularily health. I think your conclusion needs to relate to your study.
You also haven't mentioned limitations - what do we need to consider in your methodology for interpreting your data? - this is important - and future ideas. Possibly look at measuring value in an empirical way? You could do a study of conditioned place preference to measure what the ratio of resting to eliminative area would be optimal.
I think this is a good study but there needs to be effort put into presenting the introduction, method, results and discussion in a cogent way that logically informs the reader of literature, your study objectives, the methodology with more information about how you obtained the data (survey of when rooting material is replaced), your results (do a table of all descriptive data with a short text with main trends, then do stats with figures and stats in the text). Then expand your discussion talking about how the interaction of factors in pig housing can make it difficult to balance behavioural needs and health aspects of welfare such as cleanliness and make sure your statements relate to your data and that you acknowledge the shortcomings of the research.
Author Response
Reviewer 1:
L8: on 'a' resting area
Answer:
Sentence was modified accordantly (L9).
Reviewer 1:
L11: remove 'that'
Answer:
Sentence was modified (L12).
Reviewer 1:
L13: commas around as well as
Answer:
Sentence was modified (L14).
Reviewer 1:
Line 14: syntax and tense - was, not is
Answer:
Sentence was modified (L16).
Reviewer 1:
L15: The eliminating area.
Answer:
Sentence was modified accordantly (L16).
Reviewer 1:
L16: temper - quality of the resting area had an effect on usage etc. How did you measure value? Just temper this sentence a bit and return to the independent variables - you can say this simply without overstating (or consider deleting).
Answer:
Sentence was modified (L16-18).
Reviewer 1:
L26: The pen
Answer:
Sentence was modified (L28).
Reviewer 1:
L34: I find 'great importance' difficult. Possibly temper this bit.
Answer:
Sentence was changed (L35-37).
Reviewer 1:
L41: I don't understand this sentence. 'but also to? cleaning the pens'. Start new sentence after production. Ventilation sentence is out of place here.
Answer:
Sentence was changed (L43-44).
Reviewer 1:
L52: optimal? consistent - if you say optimal I want to know what that is
Answer:
Sentence was changed (L55).
Reviewer 1:
L52 The pen
Answer:
“The” was added (L55).
Reviewer 1:
L57: expand on ‘fattening period’ for those of us who aren’t in the know – 0.8m2 is not a lot of space at all (guinea pigs have more!) expand (should have access to x space for lying? Or do you mean that 0.8m2 per pig designates the overall pen size? Please clarify
Answer:
Agree, it is not enough, in EU is even less (0.65 m2; Group housing requirements lead to market turbulence - Pig Progress). 0.8 m2 for lying and resting area, with no concrete regulation, only as presented. I`ve added “in total” to be clearer (L61).
Reviewer 1:
L59: dunging, fouling, eliminating - stay consistent
Answer:
Word dunging was change with eliminating (L59-60).
Reviewer 1:
L60: put in previous paragraph as you are talking about dunging now? I'm getting lost.
Answer:
Sentence was removed (L62).
Reviewer 1:
L63: change proper to appropriate?
Answer:
Was changed (L71).
Reviewer 1:
L67: open ones? clarify parentheses too
Answer:
Explanation is added (L76).
Reviewer 1:
from L69: restructure and eliminate the question - saying the question remains I find annoying - restructure this "Encouraging usage of the elimination are could depend on location in the pen...." rather than present the info as a question answer, just state the issue.
Answer:
It was modified accordingly (L82-88).
Reviewer 1:
L73 would make a good intro to L65 onwards (except for the great important bit). I need more structure here, space, partitioning, location, air quality - present logically and concisely without questions.
Answer:
Paragraph was adjusted as suggested (L82-88).
Reviewer 1:
L77: the comfort + ref
Answer:
Ref and “the was added (L63).
Reviewer 1:
L81: citation style?
Answer:
Was removed (L90-93).
Reviewer 1:
82: expunge 'great importance' :)
Answer:
Wording was deleted (L93).
Reviewer 1:
L86: conventional pens with a concrete floor?
Answer:
Yes, info was added (L90).
Reviewer 1:
L88: wording: link between provision of bedding and rooting material and elimination behaviour requires further investigation?
Answer:
Was changed (L98).
Reviewer 1:
L93: finish the thought, thus the elimination area moves? or does elimination become ubiquitous? (I wonder if there is stimulus control at work here - the type of floor signals the usage and when the temperature rises, then stimulus controls decreases). If this is research, maybe state that as it reads as if slatted floor decreases temp by 4 degrees everywhere.
Answer:
Sentence was modified (L20-126).
Reviewer 1:
L96, remove that
Answer:
Was removed (L129).
Reviewer1:
I have reached the end of the introduction. I have learned a bit about what you are going to talk about but you haven't mentioned amount of litter in in the intro text or research that has compared straw or other rooting material (I know Pedersen et al., 2005, and Holm et al., 2008) have done some work on preferences for rooting material. There should also be expectations or hypotheses for your research based upon the literature you cite in the intro. I think you need to improve the structure a bit and provide a paragraph (ish) for each of your test variables with what research has been done for that variable. As the reader, I need to have background info.
Answer:
Except for regulation in Norway (second paragraph, crucial to understand present work presented) every paragraph presented one group of test variables like pen size and design, bedding/rooting and ambient factors. Paragraphs has been modified for greater clarity also the part about litter and rooting material (L 99-113). As this was field study, we aim broadly, especially because it was not done much until now, but more information, background, is provided in the intro.
Method:
Reviewer 1:
L11 - n=16.3?
Answer:
It is 16300 and corrected (L144).
Reviewer 1:
L112: once
Answer:
It was modified (L145).
Reviewer 1:
L113 when pig pens were at the highest density
Answer:
Was modified (L146-47)
Reviewer 1:
L115 fewer, not less
Answer:
Was modified (L148).
Reviewer 1:
L116 - how did you choose which pigs to score?
Answer:
Information was provided (L150).
Reviewer 1:
L123 - is this with litter? or without?
Answer:
Yes, with litter, info is provided now (L159).
Reviewer 1:
L124 The assessment, 'when pigs were standing'.
Answer:
Sentence is modified (L163).
Reviewer 1:
How did you come up with the scale? was it based on published research?
Answer:
Scoring system for pen and pig cleanliness was developed throughout a pilot study on commercial farms (n=10 farms). Information is provided int the text (L156).
Reviewer 1:
Data collection? but it's results. I would put the following section as a table in the results and have one paragraph with the main trends.
Answer:
I moved this section to results.
Reviewer 1:
Pen design section: simplify, and watch grammar. Could say that pen size and design was noted. I find this hard to digest - maybe refer to a table in the results for design types?
Answer:
Table was added Table 2-5 (L270-314.
Reviewer 1:
Amount of litter - not very specific - did you do interobserver reliability? Who did the measuring? How did you remain consistent? Did you take photos and assess? what is deep bedding - again, it isn't specified enough in the intro for to me to understand what this might have looked like.
Answer:
Information is provided in the text (L193). Pictures used are presented in Table 1.
Reviewer 1:
Rooting - mention assessment, not results.
Answer:
We moved the sentence regarding Class 1 and 2 here as well, as suggested.
Reviewer 1:
L147 - how? this is the method - you need to specify how you measured this variable so it is replicable.
Answer:
More information was added (201-206).
Reviewer 1:
L154 - what do you mean 'on the pen level'? 2.3 - these are results? and it's in the method? move 2.3 to results.
Answer:
Moved to results and this part deleted.
Reviewer 1:
L156 - mean doesn't really mean anything if you've provided percentages and there are only three options.
Answer:
It was removed from the text.
Reviewer 1:
Table 1 - delete - graph with your statistics presented in the text is adequate and much clearer.
Answer:
Was done accordingly. Table is deleted (L334).
Reviewer 1:
L161 - wording. You don't need the range if you have the standard error? or SD? say which.
Answer:
Data presented in the table as suggested (Table 2-5; L270-314).
Reviewer 1:
L161 - I'm not sure how you got to this calculation. 57% of pens were clean, 68% of pigs were clean - explain how you got to 69% please. Make sure the stats you provide are meaningful - I think just percentages is useful to me.
Answer:
It is stated in the text that we used only one class, class that refers to less than 10% of the body surface soiled. And within this class we got 69.4% of the pigs. I do not present anymore data for other two classes. Data moved to the table 2 (L268).
Reviewer 1:
L163 across farms? so number of pigs divided by slatted floor area? I think a table would help here. Put a column of design options, then the percentage of each. Relate to pen design section of method.
Answer:
Everything was moved to the table as suggested.
Reviewer 1:
Line 172 - bedding or litter? be consistent. what is class 1?
Answer:
Classes for amount of the litter on the solid floor were presented in the M&M. Class one is no litter. I`ve added also pictures that we used for better understanding. Class 1-4 are referred as litter amount, but class 5 is deep bedding (L191).
Reviewer 1:
174 hay
Answer:
Was changed (Table 4).
Reviewer 1:
L176 - this explanation of class needs to go earlier.
Answer:
Moved to M&M “rooting material” (L197).
Reviewer 1:
L177 - mention in method that you surveyed this info.
Answer:
It is stated under “rooting material” (L198).
Reviewer 1:
Ammonia? Air concentration?
Answer:
Ammonia.
Reviewer 1:
L194 - farm? What type of measure is this? categories? which?
Answer:
Farm ID was specified as a random effect, as there were repeated measures per farm (L251).
Reviewer 1:
L182 - concisely put - good!
Answer:
Thank you ?
Reviewer 1:
L201 on - 2dp for SE
Answer:
Was modified (L314).
Reviewer 1:
L203 - intermediate? What does this mean?
Answer:
Being in-between
Reviewer 1:
For this paragraph, put the statistics in for each comparison. This is meaningful. Figure 1 - great! Figure 2 - look at spelling. 2A is showing the pigs that scored a 3 for being clean - correct? But then how is the y axis up to 100? explain.
Answer:
Stat was moved to the text. Figure 2 modified. It is only the pigs that score 3 within the pen. 68% of pigs scored this, and because we are interesting in this score, we further used only this score. And as presented in the table 2, on average 69.4% of the pigs in field study scored that with range from 0-100 % within pen. It is only used score 3, nothing else.
Reviewer 1:
L232 - just report trends - L235 is superfluous. Put stats in the text where relevant - not in the table.
Answer:
Sentences were merged (L358-359).
Reviewer 1:
L245 hay
Answer:
Was changed (L373).
Reviewer 1:
Table 3 - make it like table 2 and put stats in the text.
Answer:
Stats was put in the text, but this are two class variables (not continue data as amount of litter), therefore presented in the figure 3 (L248).
Reviewer 1:
figure 3 - combine ABC, and DE instead of having 5 graphs with a not provided bar.
Answer:
It is combined, it is 2 class variable rooting material being provided or not as described in M&M. This is comparison if rooting material is provided or not and its effect on pen and pig cleanliness.
Reviewer 1:
Good - I'm not sure if the journal requires it but you may need to include effect sizes - these can allude to the most meaningful of your results.
Answer:
Thanks to point this out. If you think pen size, it is included, but not the whole, we divided between both areas important for the pigs and calculated space per pig, because this can provide some more information, what will happen if per pig there is more space available or less. Thus, we have calculation of slatted floor per pig and solid floor per pig. These are more relevant variables as only the size, because include both number of animals and space provided.
Reviewer 1:
Table 4 put stats in the text and have averages etc in the table.
Answer:
Was put in the text. However, this are continued variables and not the same as litter amount having 5 class variables. Here is not possible to provide averages. This is why in the fig 4 are provided relations between variables.
Reviewer 1:
Figure 4 - watch spelling. I like these.
Answer:
It was modified.
Reviewer 1:
Thoughts: you mention measuring the size of the pens but you did not analyse this data?
Answer:
Size of the pens are presented together with number of pigs in the pen. Therefore, we have slatted area per pig and solid area per pig. Even more data, more corrected according to cleanliness. It is divided within two main areas of the pen. It is described in M&M. If we account only for pen size, not taking into the account number of pigs, would we report not corrected for the number of pigs in the pen. There is no need to score cleanliness of slatted area as it is the goal to eliminate there. But it is important for solid area, but also in relation space per pig.
Reviewer 1:
L284 per
Answer:
Was modified (L416).
Reviewer 1:
L286 strongest affected by
Answer:
Strongest was deleted (L418).
Reviewer 1:
L286, presence and design of partitions
Answer:
Was modified (L420).
Reviewer 1:
L295 delete gradually (you didn't measure time)
Answer:
Was deleted (438).
Reviewer 1:
Discussion - first paragraph. Good compare contrast. overall more space (I think every captive animal would agree with you). You mention a larger resting area with enough space for all to rest is needed, but this increases ammonia concentration - so is it the size of the elimination area? Line 300 what data specifically? move this sentence.
Answer:
It was deleted and added some more info (L439-440; L445).
Reviewer 1:
L318 - does size of the eliminative area/slatted area matter? either put something about this before now as you've already set up size to be a factor. Then talk about other factors.
Answer:
I`ve added subheading, size of eliminative area and slatted area per pig were presented as first.
Reviewer 1:
L321 - always? not mentioned this before - do you mean air velocity? Include something about farm design in the method to give scope for this sentence.
Answer:
More information was provided (L469).
Reviewer 1:
L325 - tell me more (in the intro). I've alluded to this already - but the slatted floor should be signal enough but why are pigs not using it? This would be a paragraph in the intro. Such as more about L326 - this is interesting but I needed to know earlier
Answer:
It was provided such as; Pigs prefer to eliminate in corners or near pen partitions (marking), especially in open (slatted) ones (increased air velocity), but this increases the risk of attacks from neighboring pigs [11].
Reviewer 1:
L339 - good set up
Answer:
Thank you.
Reviewer 1:
Line 340 - new paragraph
Answer:
Was modified (L490).
Reviewer 1:
L342, in previous research, provision of straw has resulted in
Answer:
It is straw bedding. But have change to deep bedding for better clarity. Modified this sentence (L491).
Reviewer 1:
L344 - your study? make this clear (Table 1 with all variables and percentages would give the reader this info)
Answer:
Tables were made.
Reviewer 1:
L345 - but you just said straw resulted in higher ammonia levels? What about the frequency of replacing straw - does this have an effect?
Answer:
Deep bedding (straw) is adding more and more straw (together 47 kg per pig), every week until pigs are removed. Concept is to have high welfare, a lot of straw (Table 1, picture 5), but looks like that from the point of cleanliness and ammonia concentration is not the best option. This is why we think that combining deep bedding with additional access to eliminative area would be beneficial.
Reviewer 1:
L350 remove emotion 'rather surprising'
Answer:
Was removed (L504-505).
Reviewer 1:
L350 it seems that straw has multiple functions but also requires a lot more effort to manage. It might increase the 'value' (although I'm convinced on this word use as you didn't measure this) but it also increases ammonia.
Answer:
As mentioned earlier Straw bedding, and changed to deep bedding (class 5)
Reviewer 1:
L355 - don't assume :) expect?
Answer:
Was modified ? (L511).
Reviewer 1:
L360 - I think you could go further here (you've mentioned it in the intro but maybe make more of a point of it). What about providing more air velocity during that time in the resting area - expand on your data and what you have learned. Watch grammar here. A sprinkler system etc - but wouldn't this make one huge mess??
Answer:
More info was provided as suggested (L510-521).
Reviewer 1:
L364 - this is not what you have said earlier - larger area = higher ammonia. Watch grammar.
Answer:
Info was provided (L527).
Reviewer 1:
I'm all for providing a recommendation, but I think you should temper this and say that with advantages in some areas (e.g., semi-open partitions increasing air flow, there are disadvantages which affect usage of pen areas inappropriately causing issues of cleanliness and hygiene, and thus can negatively impact welfare, particularily health. I think your conclusion needs to relate to your study.
Answer:
Conclusion was modified. Hope this version is more acceptable.
Reviewer 1:
You also haven't mentioned limitations - what do we need to consider in your methodology for interpreting your data? - this is important - and future ideas. Possibly look at measuring value in an empirical way? You could do a study of conditioned place preference to measure what the ratio of resting to eliminative area would be optimal.
Answer:
Limitations or future work is provided in Conclusion (L540-42).
Reviewer 1:
I think this is a good study but there needs to be effort put into presenting the introduction, method, results and discussion in a cogent way that logically informs the reader of literature, your study objectives, the methodology with more information about how you obtained the data (survey of when rooting material is replaced), your results (do a table of all descriptive data with a short text with main trends, then do stats with figures and stats in the text). Then expand your discussion talking about how the interaction of factors in pig housing can make it difficult to balance behavioural needs and health aspects of welfare such as cleanliness and make sure your statements relate to your data and that you acknowledge the shortcomings of the research.
Answer:
Thank you for good comments on our manuscript. It was provided more info and details in introduction, M&M, results and discussion. Frequency of rooting material provision was part of the study, and it is described in M&M, and more info provided (L199). Straw bedding, was changed to deep litter for better clarity, and it is the concept of rearing. There is provided more straw every week and not replaced until pigs are moved. More info provided (L186). More info in discussion was added (L524-532).

Reviewer 2 Report
- To my experience, pen and pig cleanliness is quite relied on the shape of the pen. If the length/width ratio is over 1.5, pigs with an appropriate group size can clearly distinguish the areas of resting, feeding, and eliminating inside a pen at a proper space allowance level, and thus perform their behaviors in the designated areas accordingly. Is the pen shape considered in this study?
- Is influence of the management, like manure removal method and frequency, ventilation, et al., on ammonia concentration inside a pig barn taken into account? They are typically among the major factors. Otherwise, I would suggest the author to reconsider the paper title, with a clear focus on the effect of pen design, environmental enrichment provision …
- To improve the legibility, I would suggest the authors to add a typical layout of pig pens in a fattening pig house in Norway, indicating the location (front, back, sides), solid/slatted floor, and partitions…
- Evaluation on the pen and pig cleanliness is very experience based. Thus, the QA/QC on the data should be presented in the paper.
- Please double check ammonia concentration data (69.3 and 100.3 ppm) in table 1, which are much higher than that of my experience and documented files.
- Lines 301-302. Do not fully agree. Larger resting area may be beneficial to pig and pen cleanliness as the same effect of larger space allowance, while the producers typically do not provide extra lying areas for pigs. An ideal is that pigs should perform their behaviors right in the designated areas.
- Lines 309-310. Partially agree, while it is lack of supportive data here. Space allowance is determined by multiple factors. Suggest to rephrase the sentence.
- If possible, suggest to add more discussion on WHY. The results are well represented already, while it is lack of explanation relatively. For example, why increasing eliminating area is beneficial to pen cleanliness and ammonia concentration reduction?
Author Response
Reviewer 2:
To my experience, pen and pig cleanliness is quite relied on the shape of the pen. If the length/width ratio is over 1.5, pigs with an appropriate group size can clearly distinguish the areas of resting, feeding, and eliminating inside a pen at a proper space allowance level, and thus perform their behaviors in the designated areas accordingly. Is the pen shape considered in this study?
Answer:
Pen shape could give some meaningful data in an experimental set-up. Our field study was the first one, followed by experimental set-up. Results showed that pen shape did not provide any significant results, and we had 70 different pens included. But pen resting area per pig and pen eliminative area per pig did as we included this in the study. The group size is important, but not much studies have been made until now. We have another experimental set-up, including pen size, group size, and feeding strategy in the model. In that manuscript this is going to be discussed in details. But this has to be tested in experimental set up.
Reviewer 2:
Is influence of the management, like manure removal method and frequency, ventilation, et al., on ammonia concentration inside a pig barn taken into account? They are typically among the major factors. Otherwise, I would suggest the author to reconsider the paper title, with a clear focus on the effect of pen design, environmental enrichment provision …
Answer:
It is field study, with 87 different farms and systems. We collected info about manure removal and type of ventilation, but without effect of it. We have modified the tittle.
Reviewer 2:
To improve the legibility, I would suggest the authors to add a typical layout of pig pens in a fattening pig house in Norway, indicating the location (front, back, sides), solid/slatted floor, and partitions…
Answer:
We added, even though there were 70 different pen size and types included (L177).
Reviewer 2:
Evaluation on the pen and pig cleanliness is very experience based. Thus, the QA/QC on the data should be presented in the paper.
Answer:
Information was provided, partly in L150-152 and L156..
Reviewer 2:
Please double check ammonia concentration data (69.3 and 100.3 ppm) in table 1, which are much higher than that of my experience and documented files.
Answer:
These are F-values. Ammonia concentration is presented in table 5, being on average 4.1 ppm, with range 0-74 ppm.
Reviewer 2:
Lines 301-302. Do not fully agree. Larger resting area may be beneficial to pig and pen cleanliness as the same effect of larger space allowance, while the producers typically do not provide extra lying areas for pigs. An ideal is that pigs should perform their behaviors right in the designated areas.
Answer:
Indeed. According to Norwegian legislation, pigs should have large enough resting area as stated in Introduction. A lot of farmers are providing even more space than described in legislation as can be seen in the results. This information is added in discussion. In the future, there is a great need for new pens that suits pigs (L358).
Reviewer 2:
Lines 309-310. Partially agree, while it is lack of supportive data here. Space allowance is determined by multiple factors. Suggest to rephrase the sentence.
Answer:
It was rephrased (L364-365)
Reviewer 2:
If possible, suggest to add more discussion on WHY. The results are well represented already, while it is lack of explanation relatively. For example, why increasing eliminating area is beneficial to pen cleanliness and ammonia concentration reduction?
Answer:
More discussion was added (L340-45/346-349/L493-496).
Round 2
Reviewer 1 Report
A thorough proof read for grammar and syntax required. I picked up the following:
Line 16: remove ‘was’, and ‘were’. ‘Could’ not ‘should’.
Line 33: ‘pinpoint’
Line 44: “and pig health [7]. Finally, this and impair farm productivity.
Line 56: start with “Pen size…” (remove ‘meaning that’)
Line 67: ‘pr’ – is this ‘per’?
Line 68 – ‘trends’ – do you mean behaviour?
Line 85: effects
Line 88: too many t’s
Line 95: intentionally
Line 96: eliminating in the resting area and wallowing in excrement - proof ?
Line 125 – A scoring system
Line 131 – when pigs
Line 142 – even though, remove ‘a huge’.
Line 143: design (n=70), the most traditional shape is shown in figure 1.
Line 152: provided continuously
Line 161: across farms. ?
Table 2: are the SE’s for floor correct?
Line 204: orphan – title or is a bit missing?
Line 288 by 17%
Line 289: at least 0.41
Line 289: The data indicates that eliminative area needs to be large enough for several pigs to eliminate simultaneously in this designated area.
Line 292: urine can drain; what do you mean ‘different management routines’? do you mean ‘with more regular replacement and cleaning’?
Line 296 – The likelihood
Line 297: spelling increases
Line 298: How much of resting area is required per pig and whether this is proportional to the amount of solid/slatted floor per pig requires further investigation.
Line 302: remove ‘right’ – English can be a pain!
Line 311: paths – of a different substrate not related to elimination and resting?
Line 312: the amount of slatted and solid floor per pig in the current study resulted in…
Line 349: systems are
Line 350: has resulted
Line 367: evaporate ? do you mean ‘cool down’?
Line 382: The larger the eliminative area and resting area, the cleaner the pigs. Grammar
Conclusion: The larger the eliminative area and resting area, the cleaner the pigs and pens. However, larger areas resulted in higher ammonia concentration which could be mitigated by open or partly-open partitions in the eliminative area at the back of the pen. In addition, provision of large amounts of litter on the solid floor and using straw as rooting material increased pig and pen cleanliness whereas inappropriate use of the pen areas for elimination decreased cleanliness. In this study, we have identified links between housing factors that affect pig and pen cleanliness. Future research will investigate direct links and improvements to pig housing that minimize risks that negatively impact the health and overall welfare of commercially farmed pigs. – something like that?
Author Response
Dear reviewer
Thank you for sending us valuable feedback on our manuscript. We have addressed all comments systematically (see below). We hope that you will find the improved manuscript acceptable for publication in Animals (Special issue – Behaviour of Pigs in Relation to Housing Environment).